# Risk Factors for Postoperative Nausea and Vomiting After TACE: A Prospective Cohort Study

**DOI:** 10.3390/curroncol32010017

**Published:** 2024-12-28

**Authors:** Yuzhu Wang, Xin Zhou, Guoping Li, Qianzhou Lv, Xiaoyu Li, Zhiping Yan

**Affiliations:** 1Department of Pharmacy, Zhongshan Hospital, Fudan University, Shanghai 200032, China; yzwyy0506@163.com (Y.W.); lv.qianzhou@zs-hospital.sh.cn (Q.L.); 2Department of Interventional Therapy, Zhongshan Hospital, Fudan University, Shanghai 200032, China; zhouxin9406@126.com (X.Z.); li.guoping@zs-hospital.sh.cn (G.L.); 3National Clinical Research Center for Interventional Medicine, Zhongshan Hospital, Fudan University, Shanghai 200032, China; 4Shanghai Institution of Medical Imaging, Shanghai 200032, China

**Keywords:** postoperative nausea and vomiting, transarterial chemoembolization, hepatocellular carcinoma, risk factors, prospective assessment

## Abstract

**Objective:** Postoperative nausea and vomiting (PONV) was one of the common complications in patients with HCC who had undergone TACE. This study was a prospective analysis of patient data to investigate risk factors for PONV in patients after TACE. **Material and Methods:** Data were collected from 212 patients undergoing TACE in the interventional department between August 2022 and August 2023. Including: gender, age, education, BMI, operation time, concomitant underlying diseases and drugs, preoperative limosis, history of nausea and vomiting, history of kinetosis, history of smoking or drinking, and occurrence of PONV. A visual analog scale was used to measured pain. Neuropsychological status was also assessed, using the 7-item Generalized Anxiety Disorder Questionnaire (GAD-7) and the Patient Health Questionnaire-9(PHQ-9). To identify risk factors for PONV, multiple logistic regression analysis was used. The receiver operating characteristic (ROC) curve was plotted to assess the regression model. The clinical trial number did not apply in the study. **Results:** In this study, 212 out of a total of 904 patients with HCC undergoing TACE during their hospital stay were included for analysis. Among the included patients, the incidence of PONV was as high as 42% (89/212). Multiple logistic regression analysis showed that chronic gastritis (odds ratio [OR] = 10.350; *p* = 0.020), VAS (OR = 3.835; *p* = 0.003), epirubicin (OR = 26.685; *p* < 0.001), and the dosage of lipiodol (≥5 mL) (OR = 1.385; *p* < 0.001) were independent risk factors of PONV after TACE. The ROC curve demonstrated that the AUC was 0.902, the sensitivity was 84.3%, and the specificity was 87%. **Conclusions:** PONV is highly prevalent among patients with HCC after TACE. Chronic gastritis, pain, epirubicin, and the dosage of lipiodol were independent risk factors for PONV. The risk prediction model that was constructed according to the aforementioned factors demonstrated good discriminatory capacity for predicting the risk of post-TACE PONV, which can improve the recognition of medical providers, and has a good ability to prevent and treat nausea and vomiting.

## 1. Introduction

Hepatocellular carcinoma (HCC) is one of the most common forms of liver cancer. There is a notable geographical variation in the distribution of HCC. The Asia–Pacific region accounts for the majority of cases worldwide, and it causes a great burden [1].

In recent years, transarterial chemoembolization (TACE) has often been used as the preferred method of nonsurgical treatment, as its efficacy has been confirmed [2,3]. However, there are many complications associated with TACE. The most common complication is postembolization syndrome (PES). PES is defined as a syndrome that occurs between one and three days after TACE and is characterized by fever, nausea and/or vomiting, abdominal pain, etc. [4,5,6].

Although the duration of PES is self-limiting, extensive research has demonstrated that 80% to 90% of patients experience PES after TACE, and this can have a negative impact on the treatment experience of patients undergoing TACE [7,8], and it is also worth noting that nausea and vomiting are commonly observed in patients after TACE. Severe nausea and vomiting after TACE may result in electrolyte and water imbalances in patients, leading to increased discomfort, prolonged hospital stays and, ultimately, higher medical costs [4]. Concurrently, patients with HCC are frequently associated with cirrhosis and gastric esophageal varices bleeding, which ultimately results in mortality.

Irrespective of the underlying etiology, the presence of nausea and vomiting frequently results in discomfort for the patients and concern among the surgeons. The identification of parameters that are predictive of post-TACE nausea and vomiting is of significant importance. However, until now, there have been few studies investigating the factors that may predispose the occurrence of nausea and vomiting after TACE. Based on this, the objective of this study was to prospectively evaluate the incidence and risk factors of PONV in patients with HCC after TACE, so that appropriate care and intervention can be provided at an early stage.

## 2. Materials and Methods

### 2.1. Patient Enrollment and Eligibility Criteria

The data for 212 patients who underwent TACE in the department of interventional therapy in Zhongshan Hospital, Fudan University in Shanghai from August 2022 to August 2023 were collected. The study was approved by the Medical Ethics Committee of Zhongshan Hospital, Fudan University. The inclusion criteria were as follows: (i) a clinical or pathological diagnosis of primary hepatocellular carcinoma; (ii) a maximum of two TACE treatments; (iii) an age range of 18 to 80 years old; (iv) a Child–Pugh classification of liver functional status A or B and a performance status score (ECOG) of 0–1; (v) the absence of contrast agent allergy; (vi) the absence of intestinal obstruction and other primary diseases that can lead to nausea and vomiting; (vii) the willingness and ability to complete questionnaire surveys. Patients were excluded if (i) they did not meet the aforementioned inclusion criteria, lacked sufficient information regarding TACE risk factors, or had one of the following conditions: neurological diseases, mental disorders, or affective disorders (especially anxiety or depression); (ii) severe coagulopathy that cannot be corrected, cachexia or extensive distant metastasis of the tumor, or renal insufficiency. Additionally, those who refused to participate in the study and those who had communication barriers were also excluded.

### 2.2. TACE Procedure

The TACE treatment was conducted by experienced physicians in accordance with strict aseptic protocols. Following the administration of local anesthesia (2% lidocaine), the femoral artery was punctured using the Seldinger technique, and the hepatic artery was cannulated with a 4–5 Fr RH catheter. Arterial angiography confirmed the feeding artery of the liver tumors. Chemotherapy drugs (5-fluorouracil, oxaliplatin, or lobaplatin) were then administered through the catheter inserted into the tumor artery, and the tumor-feeding artery was super-selectively cannulated with a 4–5 Fr catheter or a 3F microcatheter. Subsequently, chemotherapeutics (epirubicin or pirarubicin) mixed with 3–20 mL iodized oil suspension were administered by slow injection into the tumor-feeding artery until the iodized oil suspension was deposited in the tumor bodies. Gelatin sponge particles with a diameter of 300–500 μm were then added for embolization. Finally, a repeat angiography was performed to confirm embolization of the tumor feeding artery.

### 2.3. Neurological and Neuropsychological Assessments

The participants’ demographic characteristics, including the gender, age, and level of education, were obtained through face-to-face interviews conducted by trained researchers. The body mass index (BMI) was calculated by dividing a person’s weight in kilograms (kg) by the square of their height in meters (m). In addition, the participants were asked about their smoking and drinking histories. The medical histories and laboratory tests were verified and documented. The medical histories of the subjects were confirmed and recorded in accordance with the definitions set out by the physicians, who had diagnosed and documented hypertension, diabetes, and heart disease, etc. In the study, the 7-item Generalized Anxiety Disorder Questionnaire (GAD-7) and the Patient Health Questionnaire-9 (PHQ-9) were used to evaluate the anxiety and depressive status of each participant prior to surgery [9,10].

### 2.4. Evaluation Criteria for Nausea and Vomiting

The evaluation criteria for nausea and vomiting was the Common Terminology Criteria for Adverse Events (CTCAE 5.0). PONV was assessed within the first 24 h following TACE.

### 2.5. Measurement of Pain

Pain was assessed by a visual analog scale (VAS) ranging from 0 to 10 while the participants were at rest [11].

### 2.6. Data Collection

The data relating to the sociodemographic characteristics of the patients were recorded on a structured questionnaire, such as age, gender, educational level, body mass index (BMI), living situation, exercise regime, history of PONV, smoking, drinking habits, kinetosis, hypertension, diabetes, and cardiac pathology. The remaining data was extracted from the hospital’s electronic database. The following variables were also collected from electronic medical records: concomitant underlying diseases, combination drug, operation time, dose of iodized oil, and blood biochemistry.

### 2.7. Statistical Analysis

The Statistical Package for Social Sciences version 22.0 (IBM, 187 Chicago, IL, USA) was used for all statistical analysis. The normality of the quantitative variables was assessed using the Kolmogorov–Smirnov test. Quantitative variables are presented as means and standard deviations (SDs) or medians and interquartile ranges (IQRs). The variables were then compared between the groups using either independent *t*-tests or rank-sum tests, as appropriate. Chi-squared or Fisher’s exact tests were used to compare qualitative variables, which were presented as frequencies and corresponding percentages. In addition, the independent risk factors associated with PONV were identified using logistic regression models. Variables with a *p* value of less than 0.1 in the descriptive analysis and some variables that the authors felt needed to be forced in were identified using multiple logistic regression models. The covariates that were included in the multiple logistic regression analysis of the incidence of PONV was as follows: sex (female vs. male), age, BMI, chronic gastritis (yes vs. no), kinetosis (yes vs. no), VAS, GAD-7, epirubicin (yes vs. no), lobaplatin (yes vs. no), oxycodone (yes vs. no), and dosage of lipiodol (<5 mL vs. ≥5 mL). A forward conditional approach was used to enter new terms into the logistic regression model. All *p* values were two-sided, and a *p* value of less than 0.05 was considered statistically significant. The prediction model was constructed on identified risk factors, and the receiver operating characteristic (ROC) curve was used to assess the predictive ability of the model. The area under the curve (AUC) was calculated in accordance with the established methodology. The sensitivity and specificity were calculated according to the optimal threshold, as determined by the established methodology. The statistical analyses were conducted using the Statistical Package for the Social Sciences (SPSS) version 22.0 (IBM, Chicago, IL, USA).

## 3. Results

### 3.1. Basic Information

A total of 212 patients, including 123 patients in the without PONV group and 89 patients in the PONV group, who met the inclusion criteria, were included in the study (Figure 1). As shown in Table 1, the study population consisted of 189 male and 23 female patients, representing a male-to-female ratio of 89.2% to 10.8%, respectively. The age of the subjects ranged from 25 to 80 years, with an average age of 58.6 ± 10.8 years. Two patients (0.9%) were aged ≤ 30 years, 75 (35.4%) were aged 31–55 years, and 135 (63.7%) were aged > 55 years. Limosis was performed preoperatively in 170 patients (80.2%), while 42 patients (19.8%) did not undergo this procedure. Of the total number of patients, 89 patients (42%) had a previous history of nausea and vomiting, while 123 patients (58%) had no history of such symptoms. A history of kinetosis was documented in nine patients, representing 4.2% of the total number of participants. Conversely, no cases of kinetosis were identified among the remaining 203 patients, representing 95.8% of the total participants. A total of 33 patients (15.6%) indicated that they had a history of smoking, while 179 patients (84.4%) indicated that they had never smoked. A history of alcohol consumption was reported in 30 patients (14.2%), while 182 patients (85.8%) had no such history.

As shown in Table 2, chemotherapeutic drugs were administered intraoperatively. The most commonly prescribed chemotherapeutic agents were epirubicin (51 patients, 24.1%), pirarubicin (59 patients, 27.8%), lobaplatin (97 patients, 45.8%), oxaliplatin (37 patients, 17.5%), raltitrexed (13 patients, 6.1%), and 5-FU (8 patients, 3.8%). Analgesic drugs, specifically oxycodone or morphine, were administered preoperatively in 32 patients (15.1%) and 31 patients (14.6%), respectively. The quantity of lipiodol administered intraoperatively was less than 5 mL in 93 patients (43.9%), and 5 mL or more in 119 patients (56.1%).

Postoperative nausea was observed in 89 patients, representing 42% of the total number of participants. Postoperative vomiting was observed in 57 patients (26.9%), and in the cases of postoperative vomiting, patients also experienced nausea (Table 3).

### 3.2. Incidence of Nausea and Vomiting After TACE

A total of 212 patients were included in the study. Nausea and vomiting were observed in 89 patients (42%) after TACE. The patients were subsequently classified into two groups: the first comprising those without PONV (*n* = 123) and the second, those with PONV (*n* = 89). No significant differences were observed in the demographic variables of gender, age, education, BMI, smoking, alcohol consumption, history of PONV, hypertension, diabetes, heart disease, kinetosis, ALB, ALT, AST, ALP, γ-GT, operation time, GAD-7, and PHQ-9 (*p* > 0.05, Table 1). And a rising trend was observed in the number of patients who had a diagnosis of chronic gastritis in the PONV group, in comparison to those in the without-PONV group.

However, compared to the without-PONV group, the VAS score significantly increased in the with-PONV group (*p* < 0.05, Table 1). The use of epirubicin, lobaplatin, and oxycodone were more prevalent in the group that experienced PONV than in the group that did not (*p* < 0.05, Table 2). Furthermore, significant differences were observed in the dosage of lipiodol between patients with and without PONV (*p* < 0.05, Table 2).

Multiple logistic regression analysis revealed that chronic gastritis (odds ratio [OR] = 10.350; *p* = 0.020) was an independent risk factor for PONV; in addition, VAS (OR = 3.835; *p* = 0.003), the use of epirubicin (OR = 26.685; *p* < 0.001), and the dosage of lipiodol (≥5 mL) (OR = 1.385; *p* < 0.001) were also determined to be independent risk factors for PONV(see Table 4).

### 3.3. ROC Curve Analysis

The prediction probability model based on multivariate regression was as follows [12]: prediction probability P = e^x^/1 + e^x^, where e is the natural logarithm, X = −3.091 + 2.337 (chronic gastritis) + 0.326 (VAS) + 3.284 (epirubicin) + 1.344 (dosage of lipiodol (≥5 mL)). As shown in Figure 2, ROC curve analysis was used to test the fitting effect between the predicted probability and PONV after TACE. The AUC was 0.902 (95% CI 0.857~0.947), with a *p* value less than 0.001, the optimal cut-off value was 0.713. At this point, the predictive sensitivity was 84.3%, and the specificity was 87%.

## 4. Discussion

Nausea is a visceral sensation that may eventually result in the act of vomiting. The act of vomiting is a protective reflex that involves the expulsion of gastric contents from the body through the mouth. A number of receptors have been identified as being associated with the triggering of nausea and vomiting [13,14]. TACE represents a principal therapeutic modality employed in the treatment of unresectable hepatocellular carcinoma. PONV is a prevalent and distressing complication associated with TACE, impacting patients, their families and surgeons. To date, there have been few prospective studies that have demonstrated the influential factors for predicting post-TACE nausea and vomiting in patients with HCC. Furthermore, the underlying mechanisms of post-TACE nausea and vomiting remain unclear. Based on this, the objective of our study was to prospectively evaluate risk factors associated with PONV in patients with HCC undergoing TACE, with the aim of facilitating the provision of optimal care and intervention at an early stage.

The incidence of PONV in our study cohort who received prophylactic antiemetics prior to surgery was 42% (89/212), which is relatively lower than the incidence reported in another study (52.5%) [15], but consistent with the incidence reported in another study (38.8%) [16]. Aroke et al. reported that the incidence of postoperative nausea and vomiting in surgical patients was as high as 80% in the absence of prophylactic antiemetics administered perioperatively [17]. It is therefore recommended that effective PONV rescue strategies should be employed in order to optimize postoperative recovery and improve patient satisfaction.

The results of multiple logistic regression analysis indicate that chronic gastritis, pain, epirubicin, and dosage of lipiodol were the independent risk factors for PONV after TACE in our study. Chronic gastritis was identified as an independent risk factor, a finding that was not previously reported in the literature. A nationwide multi-center survey in China revealed that 10.4% of patients with chronic gastritis experienced nauseous symptoms [18]. The occurrence of nausea or vomiting in patients with chronic gastritis may be associated with abnormalities in gastric motility. It is therefore recommended that surgeons pay closer attention to these patients with chronic gastritis and adopt a more proactive approach to their treatment.

This study is the first to demonstrate that pain is an independent risk factor for PONV after TACE. Postoperative pain is a common postoperative complication following TACE. Pain during TACE will result in the release of dopamine, epinephrine, and other transmitters, which in turn precipitate the onset of nausea and vomiting [5,19]. Additionally, in clinical practice, surgeons would attempt to provide assistance by prescribing opioid drugs if the patient complained of pain. However, nausea and vomiting are common side effects of using opioid drugs due to higher VAS scores following TACE, which may be another potential reason for PONV.

The National Comprehensive Cancer Network (NCCN) employs a four-category classification system for emetogenic antineoplastic agents, reflecting the varying degrees of emetogenic risk exhibited by different antineoplastic agents [20]. It is common practice to administer chemotherapeutic drugs, particularly during TACE, via the celiac trunk or superior mesenteric artery branches. The use of epirubicin and dosage of lipiodol were identified as independent risk factors for the development of post-PONV. The preceding study demonstrated that epirubicin and the dosage of lipiodol were independent risk factors, a finding that was consistent with the results of our study [15]. Firstly, chemotherapeutic drugs can act directly on the intestinal mucosa through the intestinal lumen, or indirectly through the blood circulation, thereby activating the receptor trigger zone. Furthermore, this results in the release of a variety of neurotransmitters, which act on the vomiting center, thereby inducing nausea and vomiting [21,22]. Secondly, the intraoperative use of chemotherapeutic drugs that enter the bloodstream and stimulate chemoreceptors in the gastrointestinal tract may be a contributing factor in the occurrence of postembolization nausea and vomiting. Finally, the results of our study demonstrated that the use of epirubicin and lobaplatin was significantly more frequent in the with-PONV group than in the without-PONV group in our univariate analysis, although lobaplatin was not identified as an independent risk factor in multiple logistic regression, possibly owing to the limited sample size or the single center study design. Therefore, further studies with larger sample sizes and a multicenter design are required to validate these findings.

The dosage of lipiodol was also identified as an independent risk factor for PONV following TACE. It was well known that the more lipiodol was used, the wider the embolization range, and the more severe the tissue ischemia and hypoxia, aseptic inflammation, and transmitters were released [23,24]. Therefore, it is now possible for surgeons to provide more precise estimates of the likelihood of PONV after TACE. And this enables them to prescribe prophylactic antiemetic agents and to deal with the postembolization nausea and vomiting promptly when it occurs [25]. In summary, the risk prediction model based on the identified risk factors has good discriminant capacity for predicting the risk prediction of PONV after TACE.

The nausea and vomiting that occurred after TACE were transient and self-limiting [26]. Patients with PONV were treated with intravenous or oral antiemetics until either mild or complete resolution of their symptoms was observed. Additionally, dietary modifications were necessary, as patients were unable to tolerate oral intake due to the presence of nausea and a lack of appetite. In our study, we predicted risk factors associated with PONV after TACE. It is crucial to gain an understanding of these risk factors in order to guarantee that patients at the highest risk are able to receive the most appropriate preventive measures prior to TACE. Due to the limited sample size and non-multicenter study design, our results may not be representative of the overall incidence of PONV after TACE. Nevertheless, our study demonstrated that the incidence of PONV after TACE remained markedly elevated, underscoring the need for greater attention to this issue. Additionally, chronic gastritis, pain, epirubicin, and dosage of lipiodol were identified as risk factors. Accordingly, a risk prediction model for PONV after TACE was established, exhibiting good predictive capacity. Such findings can provide a scientific basis for future intervention research.

## 5. Conclusions

The high incidence of nausea and vomiting after TACE should be paid more attention by surgeons and pharmacists. In addition, patients with chronic gastritis, pain, and intraoperative use of epirubicin or lipiodol in higher dosages are more prone to nausea and vomiting and should be much more cared for.

## Figures and Tables

**Figure 1 curroncol-32-00017-f001:**
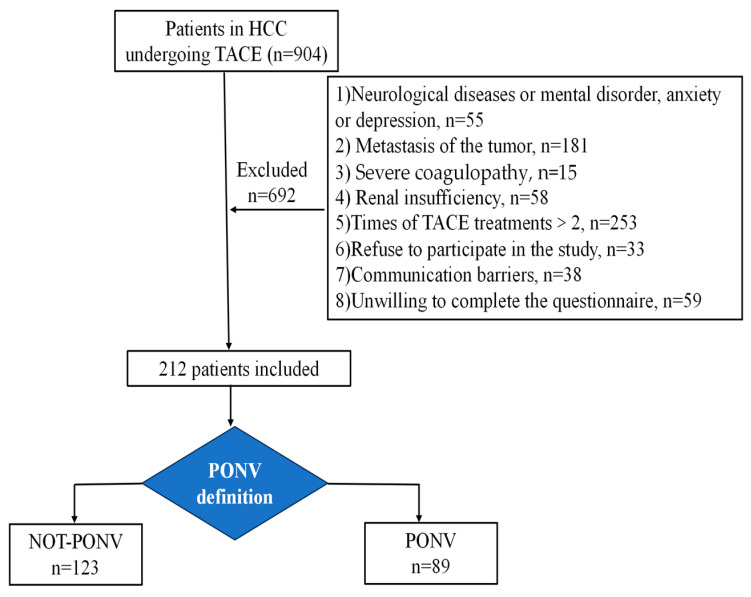
Flowchart of the study.

**Figure 2 curroncol-32-00017-f002:**
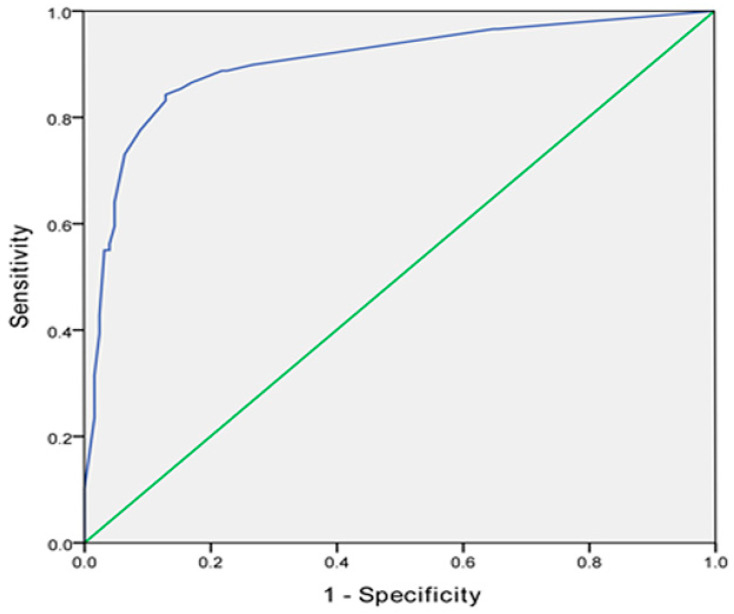
Receiver operating characteristic (ROC) curve for the model of risk factors; *p* < 0.001, AUC = 0.902 (95% CI 0.857~0.947). Logit (P) = −3.091 + 2.337 [chronic gastritis] + 0.326 [VAS] + 3.284 [epirubicin] + 1.344 [dosage of lipiodol (≥5 mL)]. Moreover, the predictive sensitivity was 84.3% and the specificity was 87%.

**Table 1 curroncol-32-00017-t001:** Demographic information and clinical characteristics of patients with and without PONV.

Demographic Information	Patients Without PONV *N* = 123	Patients With PONV *N* = 89	*p* Value
Gender, *n* (%)			0.294
Female	11 (8.9)	12 (13.5)	
Male	112 (91.1)	77 (86.5)	
Age, median (IQR)	60 (13)	57 (15)	0.128
Education, *n* (%)			0.131
Primary studies	38 (30.9)	16 (18.0)	
Secondary studies	28 (22.8)	29 (32.6)	
Senior high studies	28 (22.8)	24 (27.0)	
University studies	29 (23.5)	20 (22.4)	
BMI, mean (SD)	22.6 (3.1)	21.9 (3.4)	0.148
Smoker, *n* (%)			0.169
Never	106 (86.2)	73 (82.0)	
Occasional	8 (6.5)	13 (14.6)	
10 cigarettes daily	5 (4.1)	2 (2.2)	
≥20 cigarette daily	4 (3.3)	1 (1.2)	
Alcohol drinker, *n* (%)			0.471
Never	108 (87.8)	73 (82.0)	
Occasional	13 (10.6)	15 (16.9)	
Daily	1 (1.6)	1 (1.1)	
History of nausea and vomiting	48 (39.0)	41 (33.3)	0.305
Limosis before operation	99 (80.5)	71 (79.8)	0.898
Concomitant underlying diseases			
Hypertension, *n* (%)	40 (32.5)	28 (31.5)	0.870
Diabetes, *n* (%)	24 (19.5)	12 (13.5)	0.249
Heart disease, *n* (%)	5 (4.1)	3 (3.4)	0.793
Chronic gastritis, *n* (%)	1 (0.8)	4 (4.5)	0.081
Kinetosis, *n* (%)	3 (2.4)	6 (6.7)	0.089
Laboratory variables			
ALB (g/L) median (IQR)	39 (6)	38 (7)	0.233
ALT (U/L) median (IQR)	30 (29)	31 (23)	0.885
AST (U/L) median (IQR)	39 (29)	40 (32.5)	0.486
ALP (U/L) median (IQR)	124 (102)	138 (98)	0.192
γ-GT (U/L) median (IQR)	99 (130)	94 (159.5)	0.370
Operation time (min), median (IQR)	47 (23)	47 (20)	0.595
VAS, median (IQR)	0 (0)	6 (7)	0.003
GAD-7, median (IQR)	2 (3)	2 (2)	0.085
PHQ-9, median (IQR)	3 (3)	3 (3.5)	0.207

Data are described as mean (SD), *n* (%), or median (IQR); PONV, postoperative nausea and vomiting; BMI, body mass index; ALB, albumin; ALT, alanine transaminase; AST, aspartate aminotransferase; ALP, alkaline phosphatase; γ-GT, γ-glutamyl transpeptidase; VAS, visual analog scale; GAD-7, the 7-item Generalized Anxiety Disorder Questionnaire; PHQ-9, Patient Health Questionnaire-9.

**Table 2 curroncol-32-00017-t002:** Concomitant drugs of patients with and without PONV.

	Patients Without PONV *N* = 123	Patients With PONV *N* = 89	*p* Value
Concomitant drugs			
Entecavir, *n* (%)	71 (57.7)	54 (60.7)	0.666
Tenofovir, *n* (%)	10 (8.1)	5 (5.6)	0.481
Epirubicin, *n* (%)	21 (17.1)	30 (33.7)	0.035
Pirarubicin, *n* (%)	34 (27.6)	25 (28.1)	0.943
Lobaplatin, *n* (%)	46 (37.4)	53 (59.6)	0.047
Oxaliplatin, *n* (%)	18 (14.6)	19 (21.3)	0.204
Raltitrexed, *n* (%)	9 (7.3)	4 (4.5)	0.100
5-FU, *n* (%) Use of analgesics	3 (2.4)	5 (5.6)	0.231
Oxycodone, *n* (%)	10 (8.1)	22 (24.7)	0.008
Morphine, *n* (%)	16 (13.0)	15 (16.9)	0.434
Dosage of lipiodol, *n* (%)			0.010
<5 mL	63 (51.2)	30 (33.7)	
≥5 mL	60 (48.8)	59 (66.3)	

Data are described as *n* (%); PONV, postoperative nausea and vomiting.

**Table 3 curroncol-32-00017-t003:** Incidence of nausea and vomiting after TACE.

	Frequency	Percent
	Postoperative nausea	
No	123	58
Yes	89	42
	Postoperative vomiting	
No	155	73.1
Yes	57	26.9

**Table 4 curroncol-32-00017-t004:** Binary logistic regression of PONV after TACE.

Factors	*B*	*p* Value	OR	95% CI
Chronic gastritis	2.337	0.020	10.350	1.597~115.240
VAS	0.326	0.000	1.385	1.229~1.561
Epirubicin	3.284	0.000	26.685	10.031~70.990
Dosage of lipiodol (≥5 mL)	1.344	0.003	3.835	1.639~8.973
Constant	−3.091	0.000		

## Data Availability

The datasets used in the present study, together with the analytical methods used, are available from the corresponding author upon reasonable request.

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
