# Peer review of "Risk Factors for Postoperative Nausea and Vomiting After TACE: A Prospective Cohort Study"

_curroncol, 2024, doi:10.3390/curroncol32010017_

Round 1
Reviewer 1 Report
Comments and Suggestions for Authors
The article "Risk Factors for Postoperative Nausea and Vomiting After TACE: A Prospective Cohort Study" provides important insights into TACE and the associated risk factors for postoperative nausea and vomiting. Recommendations:
1. The introduction does not require any changes.
2. In the materials and methods section, provide more detailed information about the patients' pathology.
3. In the results section, determine whether the results are statistically significant, including a multinomial regression analysis. Highlight statistically significant findings.
4. Figures 1 and 2 should be removed.
5. The results are not statistically significant, partly due to bias, as the number of male patients is considerably higher than female patients. Additionally, the lack of a clearly defined control group makes it difficult to draw conclusions about the risk factors.
6. The discussion section should include more studies from the relevant literature.
7. Add a conclusions chapter.
Author Response
Thanks for the reviewers’ comments. These comments are all valuable and very helpful for revising and improving our paper. We have studied comments carefully and have made correction which we hope meet with approval. Revised portions are marked in highlight in the paper. The main corrections in the paper and the responses to the reviewers’ comments are as following:
Comments and Suggestions for Authors
The article "Risk Factors for Postoperative Nausea and Vomiting After TACE: A Prospective Cohort Study" provides important insights into TACE and the associated risk factors for postoperative nausea and vomiting. Recommendations:.
1)The introduction does not require any changes.
Response: Thanks for your valuable suggestion.
2) In the materials and methods section, provide more detailed information about the patients' pathology.
Response: Thanks for your valuable suggestion.The patients included in this study were those who were diagnosed with hepatocellular carcinoma in the electronic medical records. it is possible that some of the patient's had pathology that showed hepatocellular carcinoma, but some were diagnosed with hepatocellular carcinoma through clinical symptoms and imaging and did not possibly undergo puncture biopsy. One of the inclusion criteria were a clinical or pathological diagnosis of primary hepatocellular carcinoma in the study of our manuscript.
3)In the results section, determine whether the results are statistically significant, including a multinomial regression analysis. Highlight statistically significant findings.
Response: Thanks for your valuable suggestion.The results section of statistically significant findings were highlighted in paragraphs in our manuscript.
4)Figures 1 and 2 should be removed.
Response: Thanks for your valuable suggestion. Figures 1 and 2 had been removed.
5)The results are not statistically significant, partly due to bias, as the number of male patients is considerably higher than female patients. Additionally, the lack of a clearly defined control group makes it difficult to draw conclusions about the risk factors.
Response: Thanks for your valuable suggestion. According to the World Health Organization (WHO) Cancer Research Centre (IARC) in 2008, the incidence of liver cancer was ranked fifth among the most prevalent cancers worldwide, with 523,432 new cases of liver cancer in men (7.9% of the total incidence of cancers at all sites) and 226,312 cases in women (6.5%) each year. Proportion of males is higher than females. Therefore, the proportion of males in the study is relatively high. In the multivariate regression analysis of nausea and vomiting after TACE, gender, age and BMI were forced into the regression model in order to avoid bias in the analysis, even though there were no statistical significance between the two groups in the univariate regression analysis. The enrolled patients in the control group in the study did not experience nausea and vomiting after TACE according to the definition of nausea and vomiting.
6)The discussion section should include more studies from the relevant literature.
Response: Thanks for your valuable suggestion. The discussion section had been added the relevant references.
7)Add a conclusions chapter.
Response: Thanks for your valuable suggestion. Conclusions had been added.
Reviewer 2 Report
Comments and Suggestions for Authors
I appreciate the opportunity to review this manuscript by Wang et al, entitled Risk Factors for Postoperative Nausea and Vomiting After TACE: A Prospective Cohort Study. The study is overall well designed and the results are presented well. I have a few clarifications below
1. 212 out of 904 patients were included in the study. Since the majority of patients were excluded, it would be nice for the authors to include a flow chart of the selection of study population.
2. It would be good include how patients with nausea are identified and the timeline when patients experience nausea. It would be important to know whether they experience nausea as a result of TACE, or untreated post-procedure pain or analgesia.
3. How is chronic gastritis defined? is it by patient report? or does it correlate with EGD findings? are those chronic gastritis associated with H pylori? or NSAIDs use?
4. Minor grammatical issues (line 181), "...who underwent chronic gastritis..."
5. the authors found dosage of lipiodol correlates with incidence of PONV. Does higher dose of lipiodol correlate with bigger tumor size? tumor size was not a parameter included in figure 1 but could be important.
6. the authors mentioned that "it is therefore recommended that effective PONV rescue strategies should be employed in order to optimize postoperative recovery" based on previous studies. Was prophylactic antiemetics used in this study? what's the percentage of patients receiving such treatment and what drug did patient receive?
Author Response
Thanks for the reviewers’ comments. These comments are all valuable and very helpful for revising and improving our paper. We have studied comments carefully and have made correction which we hope meet with approval. Revised portions are marked in highlight in the paper. The main corrections in the paper and the responses to the reviewers’ comments are as following:
Comments and Suggestions for Authors
I appreciate the opportunity to review this manuscript by Wang et al, entitled Risk Factors for Postoperative Nausea and Vomiting After TACE: A Prospective Cohort Study. The study is overall well designed and the results are presented well. I have a few clarifications below.
1)212 out of 904 patients were included in the study. Since the majority of patients were excluded, it would be nice for the authors to include a flow chart of the selection of study population.
Response: Thanks for your valuable suggestion. A flowchart has been shown in Figure 1.
2) It would be good include how patients with nausea are identified and the timeline when patients experience nausea. It would be important to know whether they experience nausea as a result of TACE, or untreated post-procedure pain or analgesia.
Response: Thanks for your valuable suggestion.The present study investigated the incidence of nausea and vomiting within 24 hours of TACE. To ascertain whether the timeline when patients experience nausea or they experience nausea as a result of TACE, or untreated post-procedure pain or analgesia, these ideas give us provide a direction for the design of subsequent research.
3)How is chronic gastritis defined? is it by patient report? or does it correlate with EGD findings? are those chronic gastritis associated with H pylori? or NSAIDs use?
Response: Thanks for your valuable questions. The diagnosis of chronic gastritis in patients was provided by the electronic medical records. Some patients underwent gastroscopy, which confirmed the diagnosis of chronic gastritis. Those chronic gastritis may not be associated with H pylori? or NSAIDs use?.
4)Minor grammatical issues (line 181), "...who underwent chronic gastritis...".
Response: Thank you very much for discovering this error. The sentence has been revised in the manuscript. “...who had a diagnosis of chronic gastritis ”
5)the authors found dosage of lipiodol correlates with incidence of PONV. Does higher dose of lipiodol correlate with bigger tumor size? tumor size was not a parameter included in figure 1 but could be important.
Response: Thanks for your valuable questions. Higher dose of lipiodol could correlate with bigger tumor size. The size and number of tumors were not included in this study because of incomplete patient data in the electronic medical records. However, the dose of lipiodol may reflect the size of the tumour to some extent.
6)the authors mentioned that "it is therefore recommended that effective PONV rescue strategies should be employed in order to optimize postoperative recovery" based on previous studies. Was prophylactic antiemetics used in this study? what's the percentage of patients receiving such treatment and what drug did patient receive?.
Response: Thanks for your valuable questions. Prophylactic antiemetics were commonly used in patients with HCC undergoing TACE. All patients were routinely treated with antiemetics, and tolansetron was administered.
Round 2
Reviewer 1 Report
Comments and Suggestions for Authors
The authors have made the requested changes.
Reviewer 2 Report
Comments and Suggestions for Authors
All concerns addressed